# 'Our project, your problem?' A case study of the WHO's mRNA technology transfer programme in South Africa

**Matthew Herder**[1,2]*, **Ximena Benavides**[3,4]

**1** Faculty of Medicine, Department of Pharmacology, Dalhousie University, Halifax, Canada, **2** Health Justice Institute, Schulich School of Law, Dalhousie University, Halifax, Canada, **3** Program of Ethics, Politics and Economics, Yale University, New Haven, Connecticut, United States of America, **4** Information Society Project, Yale Law School, New Haven, Connecticut, United States of America

* Matthew.Herder@dal.ca

**Data Availability Statement:** In order to preserve the confidentiality of people who participated in this research, but do not wish their identities to be disclosed we are unable to share interview transcripts in their entirety. The research ethics

## Abstract

In June 2021 the World Health Organization (WHO) and the Medicines Patent Pool (MPP) launched an mRNA technology transfer programme. With a South African consortium serving as the hub, the programme aimed to increase vaccine manufacturing capacity in low- and middle-income countries (LMICs) in view of the "vaccine apartheid" that was observed during COVID-19. Following Clarke's "situational analysis," the present study assessed whether the mRNA programme differs from the approach and practices that comprise current biopharmaceutical production. Numerous documentary sources, including legal agreements underpinning the programme, funding agreements, and patent filings, were reviewed. Semi-structured interviews with 35 individuals, ranging from the programme's architects and university scientists to representatives from LMIC vaccine manufacturers taking part in the programme were also conducted. While the mRNA programme may improve the sharing of knowledge, other design features, in particular, weak conditionalities around product affordability, participants' freedom to contract with third parties, and acceptance of market-based competition, are in line with the status quo. Further, WHO and MPP's tight control over the programme evokes the dynamics that are often in play in global health, to the detriment of empowering LMIC-based manufacturers to generate mRNA products in response to local health needs.

## Introduction

In June 2021 Afrigen Biologics, a for-profit company based in Cape Town, South Africa set out to change the global landscape of biopharmaceutical production. Chosen by the World Health Organization (WHO) to serve as the hub of an mRNA technology transfer programme, Afrigen's initial task was to make an mRNA COVID-19 vaccine against SARS-CoV-2 and distribute the technology to manufacturers located in other low- and middle-income countries (LMICs). The motivation was plain: established makers of COVID-19 vaccines, especially mRNA vaccines, had largely neglected populations in LMICs [1,2]. In view of that "vaccine

board at Dalhousie University approved this research provided that participants' identities would remain confidential. Inquiries about data availability related to this project can be sent to ethics@dal.ca.

**Funding:** One author (MH) holds a Chair in Applied Public Health, funded by the Canadian Institutes of Health Research (CIHR) and Public Health Agency of Canada (PHAC). This Chair carries a salary award as well as funding for research activities. However, neither CIHR nor PHAC played any role whatsoever in the design of the present study, data collection, analysis or writing process.

**Competing interests:** We have read the journal's policy and the authors of this manuscript have the following competing interests: Matthew Herder was a member of the Patented Medicine Prices Review Board (PMPRB), Canada's national drug pricing regulator, and received honoraria for his public service, June 2018 – February 2023. The PMPRB had no role whatsoever in the design or conduct of this research. Ximena Benavides worked for GAVI - The Vaccine Alliance, COVAX Facility, from May to October of 2021, as a Yale Institute for Global Health fellow.

apartheid,"[3] building capacity to make vaccines locally for local populations became imperative. The WHO turned to a model of knowledge-sharing that was previously deployed in response to concerns that the global influenza virus sharing network was under-serving people in LMICS [4–6]. Another Geneva-based organization, the Medicines Patent Pool (MPP), was charged with managing the mRNA programme's fundraising and legal needs.

Within six months Afrigen succeeded in producing its own mRNA COVID-19 candidate, "AfriVac 2121 [7].". The programme has the potential to be transformative as a model of vaccine production [8], encompassing both upstream research and development (R&D) and 'end-to-end' vaccine manufacturing. Still, the initiative faces several risks, including precarious levels of funding, the looming threat of patent litigation by established mRNA vaccine manufacturers, and a range of governance issues that have complicated the work of an organization created out of dire need—all the while trying to develop the technical capacity to produce high-quality mRNA-based technologies that protect against not only COVID-19 but also tuberculosis (TB), malaria, human immunodeficiency virus (HIV), and other diseases that disproportionately afflict people in LMICs.

We set out to study, using qualitative research methods, to what extent the WHO/MPP-managed mRNA programme differs from the approach and practices that comprise current biopharmaceutical production. We describe the key features of the status quo as a basis for comparison with the mRNA programme under our findings below. Combining insights from documentary sources, including the legal architecture underpinning the programme, patent filings related to mRNA products, and data from semi-structured interviews with 35 individuals involved in the programme, we find that the design of this initiative is largely in line with dominant approaches to vaccine production, steeped in the neocolonial dynamics that are often in play in the sphere of global health [9–14], and at risk of failing to enhance equitable access even if it ultimately succeeds in one day making mRNA vaccines.

## Methods

### A 'situational analysis' of the mRNA programme amidst global power imbalances

Our research followed a "situational analysis" approach—a form of grounded theory, which develops theories through observations and multiple sources of data [15]. Under situational analysis, data collection and analysis occur in parallel, requiring constant comparison between new sources of data and the preliminary, but evolving, analysis. We describe the multiple sources of data incorporated into our situational analysis below, which has been applied by social scientists to gain insight into complex systems, comprising a variety of actors with diverging interests [16,17]. At the same time, we were cognizant of the power imbalances that afflict global health from the study's inception [13,18–21]. Attention to power differences among the variety of actors and institutions engaged in the mRNA programme, and the multiple drivers of power imbalances, was central to our data collection and analysis.

### Document analysis

Multiple types of documents were analyzed by both researchers; a few minor inconsistencies in interpretation occurred but were resolved through discussion. The first type of document was a range of legal documents that codify the relationships between different actors in the mRNA programme, which, pursuant to a memorandum of understanding, the WHO tasked MPP with drafting and implementing. These "programme agreements [22]" are in place between MPP and the three principal members of the South African "consortium", that is,

Afrigen, another Cape Town-based vaccine manufacturer called Biovac, and the South African Medical Research Council (SAMRC). The "technology transfer template agreement," which served as the basis for negotiations with LMIC manufacturer partners to the hub, as well as the finalized agreements between MPP and thirteen of the fifteen programme "partners" that have signed a technology transfer agreement, all of which are publicly available from MPP's website (accessed: March 30, 2024), were also analyzed. (Only Bio-Manguinhos (Brazil) and BiovaX (Kenya) have not signed such an agreement). Additionally, research agreements shared by interview participants were analyzed, including a funding agreement between scientists at the University of Cape Town and the SAMRC, a research collaboration agreement between the United States' (US) National Institute of Allergy and Infectious Diseases (NIAID, a component of the National Institutes of Health (NIH)) and Afrigen, as well as sample clauses from Afrigen's collaboration agreements with entities outside the programme. Powerpoint presentations and other information shared at the programme's inaugural meeting held in Cape Town, April 17–21, 2023, as well as a regional meeting in Bangkok, Thailand, October 31 –November 1, 2023 were also incorporated into the study. To gain insight into the relationship between countries sponsoring the programme and WHO/MPP, an access to information (ATI) request was filed with the Canadian government, which is the second highest funder of the mRNA programme. Our ATI request yielded 153 pages of correspondence, agreements, and other documentation that we incorporated into our analysis. (*see* S1 Letter and S1 Document for further details about our request and the corresponding disclosure package) Finally, a dataset of patent applications as well as withdrawn and granted patents, compiled and made publicly available by MPP [23] was analyzed to understand the evolving patent landscape related to mRNA technologies in South Africa and other LMICs tied to the programme.

## Semi-structured interviews

We used a purposive sampling strategy, contacting individuals that hold leadership positions within their respective organizations or who have relevant experience, for example, in a relevant scientific field. Within the consortium (n = 12), we interviewed executives of Afrigen (3) and Biovac (1), as well as officials from SAMRC and other parts of the South African government (3), and university-based scientists (5). We also interviewed WHO (3) and MPP (4) officials, which we refer to as the 'programme's architects' (n = 7), and representatives from vaccine manufacturers based in Argentina (2), Brazil (2), Serbia (1), India (1), Bangladesh (2), and another LMIC (2), which are now described as programme partners (n = 10). Finally, we interviewed scientists from the global North and other outside experts, businesses, and organizations (n = 6) that have supported or taken part in the programme in some fashion or work in the field of epidemic preparedness. Only one individual (of 36 that we contacted) declined to participate in an interview. The majority of interview participants (29 of 35 that chose to participate) agreed to be interviewed 'on the record', allowing quotations to be attributed to them by name. (**Table 1**) One researcher (MH) traveled to Geneva, Cape Town, Chicago, and Bangkok to recruit and run interviews in person (n = 16). Nineteen interviews took place virtually and usually involved both researchers (MH, XB).

## Research ethics statement

We received ethics approval to conduct this study from Dalhousie University's Social Sciences and Humanities Research Ethics Board (REB# 2022–6457) and Yale University's Institutional Review Board (IRB#2000034524). After discussing the purpose, benefits and risks associated with our research, all individuals we interviewed provided verbal consent to participate in the

**Table 1. Overview of research interview participants.**

|  | Organization | Participant (Initials Used) |
|---|---|---|
| *South African Consortium* (n = 12) | Afrigen | Petro Terblanche (PT) |
|  | Afrigen | Caryn Fenner (CF) |
|  | Afrigen | Amin Khan (AK) |
|  | Biovac | Patrick Tippoo (PTi) |
|  | South Africa Medical Research Council | Richard Gordon (RG) |
|  | Government Agency (South Africa) | *Anonymous (XX) |
|  | Department of Science and Innovation (South Africa) | Glaudina Loots (GL) |
|  | University of Cape Town | Anna-Lise Williamson (ALW) |
|  | University of Cape Town | Ed Rybicki (ER) |
|  | University researcher (South Africa) | *Anonymous (XY) |
|  | University of Witwatersrand | Patrick Arbuthnot (PA) |
|  | University of Witwatersrand | Charles de Koning (CdK) |
| *Architects of the Programme* (n = 7) | World Health Organization | Martin Friede (MF) |
|  | World Health Organization | Claudia Nannei (CN) |
|  | World Health Organization (formerly MPP) | Erika Dueñas Loayza (EDL) |
|  | Medicines Patent Pool (formerly WHO) | Marie-Paule Kieny (MPK) |
|  | Medicines Patent Pool | Charles Gore (CG) |
|  | Medicines Patent Pool | Chan Park (CP) |
|  | Medicines Patent Pool and Public Citizen | Peter Maybarduk |
| *Spokes/Partners* (n = 10) | Bio-Manguinhos | Sotiris Missailidis (SM) |
|  | Bio-Manguinhos | Patricia Neves (PN) |
|  | Synergium Biotech | German Sanchez Alberti (GSA) |
|  | Synergium Biotech | Fernando Lobos (FL) |
|  | Biological E. Limited | Vikram Paradkar (VP) |
|  | Torlak Institute | Luka Dragcevic (LD) |
|  | LMIC Partner | *Anonymous (XZ) |
|  | LMIC Partner | *Anonymous (XA) |
|  | Incepta | Mohammad Mainul Ahasan (MMA) |
|  | Incepta | Mahbubul Karim (MK) |
| *Global North Scientists & Other Outside Experts* (n = 6) | Government Agency (United States) | *Anonymous (XB) |
|  | University researcher (United States) | *Anonymous (XC) |
|  | University researcher (United Kingdom) | Sarah Gilbert (SG) |
|  | Global Health Innovation Alliance Accelerator | Julie Barnes-Weise (JBW) |
|  | Coalition for Epidemic Preparedness and Innovation | Emma Wheatley (EW) |
|  | Quantoom Biosciences | Jose Castillo (JC) |

*Notes*: (1) We do not describe the specific roles of each participant within their respective organizations in order to preserve participant confidentiality for those who wish to remain anonymous. For participants who agreed to be identified, we describe their roles in the main body of the article when quoting from interview data. (2) The organizational affiliation provided for each named participant does not capture nuances in the roles or experiences of each participant. For example, Peter Maybarduk is a member of MPP's Governance Board. However, he is also an employee of Public Citizen, a civil society organization based in the United States. Similarly, two other participants (Erika Dueñas Loayza and Marie-Paule Kieny) have worked with both WHO and MPP.

study at the outset of each interview. Consent was thus recorded as part of each interview transcript. All interviews occurred between February 2023 and January 2024.

### Data coding and analysis

Consistent with situational analysis, data collection and analysis occurred in parallel. We created memos summarizing key exchanges or text, interpreting both interview and documentary data to identify lines of inquiry and points to follow up during future interviews. MH and XB generated a list of concept areas, in turn, developing a set of situational maps to define relationships between all the entities involved in the mRNA programme as well as key dynamics (e.g., influence of funding organizations; competing institutional priorities) that are often operative in the field of global health and access to medicines. We followed a constant comparative method throughout our research process, and met regularly to discuss uncertainties, unresolved questions, and points of divergence among interview participants.

### Review by independent equity advisory committee

Our research process, data analysis, and preliminary findings were developed in consultation with an Independent Equity Advisory Committee (IEAC). Comprised of six members with diverse expertise in clinical trials, global health policy, access to medicines, and bioethics, the IEAC has extensive experience working with or inside organizations, such as WHO, the South Centre, Universities Allied for Essential Medicines, Médecins Sans Frontières, and the Health Justice Initiative. While the IEAC had no direct involvement in data collection, access to interview data, or control over our analysis, it played an essential role in helping to identify potential participants and critically appraising our preliminary findings and, at bottom, ensuring that our approach was attentive to the larger social and political context in which our research is situated.

## Findings

We first examine the mRNA programme's origins (2020–2021) and then compare its design to the four paradigmatic features of global biopharmaceutical production, which we abstracted from a review of literature and evidence from numerous scholarly disciplines; namely, 1) weak conditionalities attached to publicly funded science; 2) secret, transactional R&D partnerships; 3) a high degree of financialization; and, 4) market-based governance. Below, we elaborate upon, and juxtapose these four features against, our findings about the mRNA programme following an examination of the political context and policy choices that were made early on during the pandemic yet, as we show, continue to constrain the programme's approach and practices.

### Politicized origins: Building the mRNA technology transfer programme

Foreseeing access challenges from the start of the pandemic [2,24,25], WHO became home to several attempts to improve access to COVID-19 vaccines and other needed interventions in LMICs. Each differed markedly in terms of their approach to mitigating access challenges and the actors involved. The first, the "Access to COVID-19 Tools Accelerator" (ACT-A), was launched in April 2020 by a mix of public and private actors, including WHO, the government of France, the European Commission, the Bill and Melinda Gates Foundation, and three biopharmaceutical industry associations [25]. The vaccine-focused arm of ACT-A, COVID-19 Vaccines Global Access or "COVAX" (governed by Gavi, the Vaccine Alliance, the Coalition for Epidemic Preparedness Innovations (CEPI), and WHO), was intended to procure vaccines

for LMICs by leveraging the collective purchasing power of high-income countries (HICs). With HICs prioritizing domestic populations at the expense of equitable global distribution, however, COVAX's charitable approach failed to secure vaccines for LMICs [26–28]. A second initiative, the COVID-19 Technology Access Pool (C-TAP), created by WHO, the government of Costa Rica, and other member states, followed in May 2020 [29]. In contrast to ACT-A's charity-based approach, C-TAP sought to distribute control of the intellectual property (IP), data, and knowledge related to COVID-19 interventions. Pooling a variety of technologies through voluntary licenses, vaccine and other product manufacturers could in-license technologies to address population needs in LMICs [30] rather than depending on vaccine donations from HICs—a move applauded by civil society but fiercely contested by industry, its allies, and the Gates Foundation [31,32].

Meanwhile, individuals inside and adjacent to WHO began crafting a third proposal, predicated on building capacity to manufacture vaccines *in* LMICs *for* LMICs. Martin Friede, the WHO's lead coordinator for vaccine research, and Marie-Paule Kieny, the Chair of MPP's Governance Board and formerly an Assistant Director-General at WHO were especially influential. Drawing upon a "hub and spoke" model of vaccine manufacturing that WHO deployed once before [5,33,34], they envisioned a centralized knowledge sharing system with a view to enhancing local vaccine production capacity in LMICs. Friede recalls how they arrived at this model in the context of influenza vaccines:

> [I]t was very easy finding experts in terms of the vaccines because the world is full of retired people used to making influenza vaccines, but [. . .] we realized these were generally quite old gentlemen and they got very tired going around the world teaching the same process at each facility [. . .]. And this is when the concept was born of us creating a central hub, again, a corporate direction of interest. (MF)

Several crucial questions about the design of the model, in the context of COVID-19, nevertheless remained: How would it fit within WHO and the organization's other newly launched COVID-19 access initiatives? Who would oversee its operations? Would there be one central hub or several spread across different regions? What vaccine platform(s) should command its focus for technology transfer purposes?

According to the lead of WHO's IP Unit, Erika Dueñas Loayza, the original plan was to embed the COVID-19 hub within C-TAP. On behalf of WHO, Loayza's team was actively seeking voluntary licenses from COVID-19 vaccine manufacturers.(EDL) Any new IP generated by the hub or its spokes would in turn become part of C-TAP's pool, thus distributing control to LMIC-based manufacturers as their productive capacity increased. However, as industry opposition to C-TAP grew because of the threat that it posed to IP-holders' control over COVID-19 interventions [32], then-WHO assistant director general (ADG) of access to medicines and health products, Dr. Mariângela Simão, and then-WHO Chief Scientist Dr. Soumya Swaminathan opted to "move the mRNA [programme] away from C-TAP to the ACT-[Accelerator]"—an outcome that MPP's Kieny also favoured.(EDL) Although CEPI was the nominal lead for the "development and manufacturing" workstream within COVAX [25], it was the Kieny-led MPP that would later assume, in concert with WHO, responsibility for the design, day-to-day oversight, and fundraising for the hub.(CG)

Once positioned inside ACT-A, WHO issued a call for expressions of interest for "one or more" technology transfer hubs in April 2021 [35]. Afrigen's chief executive Petro Terblanche remembers recognizing the opportunity: "We are small, but we know tech transfer."(PT) Terblanche's strategy of assembling a "consortium" together with Biovac and SAMRC for the

WHO application proved wise. Friede describes the decision-making process inside WHO, which culminated in the selection of the Afrigen-led proposal on July 21, 2021:

> [T]he WHO's PDVAC, which is the production and development of vaccines advisory committee, decided that mRNA was the first platform to go for first because of its flexibility, potentially lower cost, and speed with which you could look at different antigens and whether they work or not. [. . .] Then WHO put out a call for expressions of interest to be the hub initially, and a number of companies applied. South Africa came with a consortium, which is the only one that did come with a consortium, consisting of Afrigen as the hub, Biovac as the first spoke and the South African Medical Research Council as the research institute to feed into potentially new pathogens, potentially second-generation technologies and so on. And so that was attractive because they came as a consortium, and clearly also the fact that it was in Africa was attractive to them because Africa was the standout continent of inequity and access.(MF)

Brazil's Bio-Manguinhos, a non-profit, state-owned company that is part of the Oswaldo Cruz Foundation, submitted a competing bid. Their proposal contemplated building 'end-to-end' mRNA manufacturing capacity, that is, the complete production process—from antigen design to producing the drug substance, drug product, and the fill and finish phase of inputting doses into vials—and then transferring the know-how from one LMIC manufacturer to another.(PN) Sotiris Missailidis, then head of vaccine innovation at Bio-Manguinhos, details how things shifted in the months that followed the Afrigen announcement in June 2021:

> Africa was announced first, but I think it's important to say that, in the beginning, at least, what we had understood [. . .] was that the model was going to be a decentralized model. So there were going to be two hubs in Africa [and] there were going to be two hubs, regional hubs in Latin America. There were going to be two regional hubs in Asia. [. . .] And each of them then would have spokes, potentially, that would be partners that had an interest in producing and accepting the technology. So we applied to be original hub. We didn't apply to be a spoke, ever. And we got selected. [. . .] What I didn't know was that, at some stage, [. . .] there was a decision taken from WHO or whoever, that as there was increasing political and financial pressure, many people wanted to come in. [. . .] So the decision was taken to have one central hub and everybody else would be spokes.(SM)

It remains unclear why the change in plans occurred. Bio-Manguinhos learned of their 'spoke status' when they visited Afrigen in April 2022—six months after WHO indicated they would be a regional hub.(SM,PN) [36] The minutes from WHO's PDVAC meetings show that multiple hubs were still being contemplated as late as November 2021 [37]. According to Patricia Neves, project manager for Bio-Manguinhos' center for vaccines using mRNA, MPP officials queried "why are you here [in Cape Town visiting Afrigen] if you are developing your own technology?"(PN) At the time, MPP was focused on securing a voluntary license from Moderna or another more established mRNA manufacturer even though WHO had previously tried and failed to secure such a license.(EDL) MPP's track record shows that it adheres to the norms of market-based competition and contract-based solutions even though voluntary licenses frequently exclude countries with strong manufacturing capacity, such as Brazil [38,39]. Perhaps this organizational philosophy explains why MPP appeared to be unsupportive of Bio-Manguinhos' plan to establish end-to-end manufacturing capacity, at least in early 2022. Yet, Missailidis explains why mastering every step of the production process is critical to national health security:

We don't do fill and finish. We need to have all the technology transferred up to. . . Well in the traditional vaccines, the master cell bank and everything. And we need to be able to produce [the active pharmaceutical ingredient] 100% properly. This is a condition for any tech transfer we've ever done [. . .] because of guaranteed national production in Brazil. [. . .] [H]istory shows that when the epidemics or pandemics or whatever else, you can't guarantee that you have the vaccine when you want it. [. . .] So, when we spoke, for example, to Moderna for COVID, didn't even speak with us. Pfizer did, but they were not eager to do a tech transfer, they wanted to do fill and finish. Which we said 'Look, you know we're not doing that. This is not our motto. That's not how we work.'(SM)

Two years on, the mRNA programme continues to evolve. The programme currently encompasses a diverse array of actors, including the South African consortium and fourteen other LMIC-based spokes (**Fig 1**), which are now referred to as 'partners' because of the negative connotations of the term 'spoke'.(CG) The programme's architects have also come around to the idea of creating the end-to-end manufacturing capacity first espoused by Bio-Manguinhos and echoed by Afrigen not long after it became the hub. At a meeting in Bangkok in the fall of 2023, WHO and MPP officials outlined potential sub-consortia—engaging partners both inside and outside of the programme—focused on R&D around pathogens of shared, regional interest. In this way, Bio-Manguinhos or other manufacturers that assume the lead for a particular sub-consortium might become *de facto* hubs for a given target.(CN) Expanding the programme's focus upstream is also seen as crucial to its overall sustainability given that

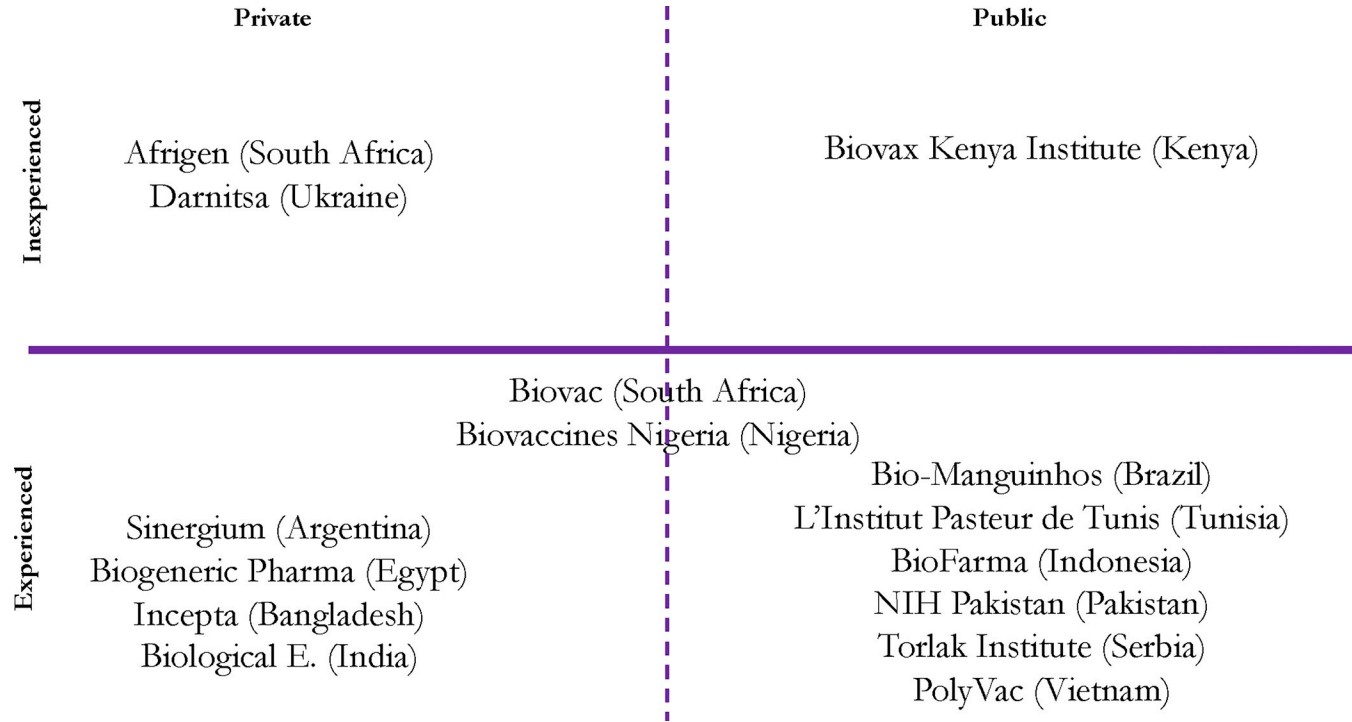

**Fig 1. Ownership and experience of mRNA hub and spokes participating in the mRNA programme.** *Notes*: (1) Two participating manufacturers, Biovac (South Africa) and Biovaccines Nigeria, are shown on the boundary between public and private ownership because each entity is a public-private partnership. All other entities depicted in the figure are state-controlled enterprises (i.e., publicly owned) or private companies. (2) For the purposes of this figure, the term 'experienced' refers to entities that have produced at least one vaccine that has been licensed for clinical use. Several manufacturers that fall into the 'experienced' half of the figure have produced more than one vaccine. The entities in the 'inexperienced' sphere have not yet fully developed a vaccine; however, some have generated sales through other products.

demand for COVID-19 vaccines is now limited.(PT,CG,AK, MF, CN) Yet, as we show in the sections that follow, a number of choices made by its architects about what commitments participation in the programme entails, what kinds of support should be provided to Afrigen and others, and how the programme is governed, may limit the programme's potential as a collaborative effort to improve equitable access to mRNA interventions in LMICs. Our analysis reveals that the programme's relatively weak commitments to access and affordability, preservation of companies' respective freedom to contract, consolidation of control by powerful actors in Geneva, and deference to the market as the ultimate arbiter of which entities will survive, both resembles the status quo and risks fragmentation within the programme, to the potential detriment of equitable access in LMICs.

## Critical inputs from publicly funded science, weak conditionalities & measured charity

The first defining feature of biopharmaceutical production concerns the limited *quid pro quo* that the public sector receives in exchange for supplying private actors with financing, biopharmaceutical R&D, and product leads. Despite significant government investments in, and publicly funded researchers' extensive contributions to the development of vaccines and other products [40–45], **weak conditionalities** tend to be attached to government and philanthropic funding of biopharmaceutical R&D. Generally, public funding (whether in the form of research grants, collaborative research agreements, or advance procurement agreements) stipulates an obligation of data transparency (e.g., publishing studies). Also, the right of university scientists to continue to conduct research with the technology in question is usually included in IP agreements. But clauses that stipulate where manufacturing should occur, when and where products can be distributed, or how resulting goods are to be priced, are not standard [46–49]. Instead, conventional wisdom is to grant maximum discretion to recipients of public funding, including universities and government laboratories, as well as private actors about how to commercialize biopharmaceuticals. Under this logic, informed by the dominant approach that maximization of profits encourages innovation, the state's role is not to shape—but simply subsidize—biopharmaceutical innovation [47].

Consistent with most early-stage biopharmaceutical R&D, the funding for the mRNA programme comes solely from governmental sources. Charged with the responsibility of fundraising, MPP secured financial commitments from France, the European Commission, Germany, Norway, Belgium, and Canada alongside the government of South Africa and the African Union [50]. To date, these donors have committed USD 117 million to the programme (with USD 89 million received so far (CN), the majority (73%) of which has been allocated to the consortium, including Afrigen, with the remainder (27%) supporting LMIC partners. According to MPP, which holds the bulk of the funds in Geneva, the USD 117 million is "seed money." By 2026 the programme is expected to be "self-sustaining [50]." Still, MPP is continuing to seek additional funding, in particular, from the US government, which, in contrast to the WHO's influenza technology transfer hub [51], has yet to offer any direct financial support for the mRNA programme.

Countries donating funds—or contemplating doing so—have shaped the programme in multiple ways. Germany, for example, earmarked a portion of its funding for a staff position at the hub. With only German or French nationals deemed eligible for the role by the funder, however, Afrigen was unable to fill the position.(PT) The government records obtained through an access to information request reveal that Canada, the second largest donor country, has stipulated that its funding be allocated to the hub in Cape Town and four select countries hosting manufacturers participating in the programme: Senegal, Nigeria, Kenya, and

Bangladesh [52]. Further, according to one interview participant, while HICs are supportive of transferring technology to LMICs, they would prefer that such transfers do not extend to the more upstream inputs into mRNA vaccine production, including novel LNPs and antigens. Nevertheless, researchers at a number of publicly funded institutions located in South Africa as well as others abroad, including the University of Witwatersrand (Wits), the University of Cape Town and the North-West University, the University of Pennsylvania, as well as the US' NIH/NIAID, have already made substantial contributions to various aspects of mRNA product development at Afrigen.(CF,PT,PA,CdK,XB)

At the start of the manufacturing process the aim is developing an 'antigen' that will provoke an immune response, conferring protection against a given pathogen. For Afrigen's Afri-Vac 2121, the lab of Patrick Arbuthnot at Wits drew upon information already in the public domain to design a plasmid, a circular piece of DNA which can be propagated efficiently in bacteria and then prepared in larger amounts to use as a template, and shared it with Afrigen. (PA) NIAID's Vaccine Research Center (VRC) similarly contributed to the plasmid construction and purification steps in line with current "good manufacturing practices" (cGMP) standards set by regulatory authorities.(XB,CF) After entering into a Research Collaboration Agreement with Afrigen in March 2022, the VRC shared its knowledge and hosted Afrigen scientists for onsite training [53,54]. Demonstrating the value of being part of a consortium, Afrigen will pull in more contributions from publicly funded researchers at Wits, the University of Cape Town, and other South African universities, as it increases its focus on the development of second-generation technologies, such as novel lipid nanoparticles (LNPs),(CdK) and new disease targets like TB, malaria, and HIV.

The critical question is whether the funding that has been secured for the programme and supporting the development of these second-generation mRNA technologies has been leveraged into a shared set of commitments geared towards improving equitable access. The relationships among the actors involved in the mRNA programme are defined by a set of legal agreements crafted by MPP. (*see* S1 Table) Under the technology transfer template agreement and all but one finalized technology transfer agreement involving MPP and LMIC partners, the latter are granted a "non-exclusive, royalty-free, non-sublicensable, non-transferable, irrevocable, fully paid-up, royalty-free licence" to the technology as well as any rights held by Afrigen and the Biovac "to make, or have made, use, offer for sale, sell, have sold, export or import" in their respective territories and other LMICs (as defined by the World Bank) [22]. In return, LMIC partners must grant to MPP upfront a "worldwide" non-exclusive, royalty-free license to "practice and have practiced the data and the Inventions for the purposes of fulfilling its mission to facilitate the development and equitable access of health technologies" that is "non-transferable, but sub-licensable." As WHO's Friede explains, the programme is akin to an IP sharing club comprised of the South African consortium as well as the thirteen other LMIC manufacturers that have signed an agreement to date:

> [T]he key objective here is that for us, open means open for LMICs. It does not mean open for [HICs]. So, if Wits can generate some revenue providing a license for use and sale within [the] US, Canada, Europe, Australia, good for them, on condition that for all of the LMICs, there is a fully paid-up, royalty-free license available.(MF)

The programme's pooled, multilateral approach to knowledge production is rare in the biopharmaceutical sector. MPP's head legal counsel, Chan Park, notes that this deviation from standard practice stems from the fact that MPP was in a fundamentally different position compared to when it is attempting to secure a voluntary license from a multinational pharmaceutical company to an existing therapy:

When we're negotiating with big pharma on a drug that they have already developed and are commercializing, our leverage is far lower. We can ask nicely for it and if they say no, we can ask again, and if they say no again, we just have to live with it. But here we're building it from the ground up. We're providing the funding, and so we can say, 'This is a project for [LMICs] and that's it.'(CP)

Still, there are a number of notable incongruities embedded in the programme's underlying legal architecture, which run the risk of fragmenting the larger, collective enterprise of improving equitable access to mRNA products in LMICs. To start, some of the partners have yet to sign on. According to Bio-Manguinhos' Missailidis, the Brazilian manufacturer cannot sign such an agreement because of its pre-existing, exclusive technology transfer agreement with AstraZeneca.(SM) His colleague leading Bio-Manguinhos' mRNA vaccine project, Patricia Neves, also intimates that the idea that technology developed by Bio-Manguinhos, using funding from the Brazillian government (as opposed to funding from the mRNA programme) would flow to manufacturers from participating LMICs, which in some cases, are for-profit commercial entities, without anything in return is an "injustice."(PN)

The issue of royalties also proved to be a sticking point within the South African consortium. According to a South African government official, a lot of back and forth with MPP was required:

[I]f somebody has spent 20 years developing a piece of IP, it's really hard for them to say take it and go and do what you like with it. And a manufacturer can make a markup of 15%, but I'm going to get nothing from it. So that to us was a disconnect that we had quite a lot of discussion around. [. . .] And we can't just have somebody else outside the country making money offered, but we have to balance that with affordability and access. And that's the balance we're constantly struggling to achieve.(XX)

An unevenness between LMIC partners and the South African university laboratories funded by the SAMRC, where the former must share their IP royalty-free and the latter may expect a return, is thus embedded in the programme. (CP,XX) None of the executed technology transfer agreements between MPP and LMIC partners state this; on the contrary, the license granted from MPP to partners is framed as "royalty free." However, the Grant Agreement with SAMRC grants MPP a "non-exclusive, transferable, sublicensable, irrevocable, worldwide, license to practice and have practiced the data and Inventions for the purposes of fulfilling its mission to facilitate the development and affordable and equitable access of mRNA technologies in low- and middle-income countries (as defined by the World Bank), which license may include a royalty sacrifice." (*see* S1 Table) Thus, inventions patented by SAMRC-funded researchers, including second-generation mRNA technologies such as the novel LNP, may be rewarded with royalties whereas new IP generated by partners using mRNA technology will not.

A second IP-related incongruity in the programme's legal architecture concerns the territorial limitations imposed upon IP licenses among different actors involved in the mRNA programme. As the central intermediary, MPP is granted a "worldwide" license to IP that is generated by both members of the South African consortium as well as LMIC partners. In turn, partners (with the exception of Indonesia's BioFarma) are entitled to receive IP via MPP but only for use, sale, export or import within LMICs. For its part, BioFarma managed to negotiate a "worldwide, non-sublicensable" license to the IP it receives from MPP; it can therefore use, sell, and export such IP globally, but it cannot sub-license it to other entities in LMICs or HICs. That flexibility of licensing IP they generated to companies based in HICs only extends

to members of the South African consortium (excluding Biovac as one of the partners). (*see* S1 Table)

Notwithstanding the leverage the programme's funding conferred, MPP also stopped short of requiring that resulting mRNA products be priced affordably for populations in need outside of a "Public Health Emergency of International Concern" (PHEIC). If an mRNA product developed by one or more LMIC partners targets a PHEIC, they cannot charge more than the cost of production plus a twenty-percent mark-up [22]. (*see* S1 Table) However, none of the pathogens being targeted by the programme's partners—from TB to respiratory syncytial virus (RSV), malaria, and other infectious diseases—are currently designated as a PHEIC. Thus, consistent with industry norms, the mRNA programme does not contractually constrain partners' pricing decisions. Rather, the assumption is that by targeting LMIC markets, the price of the final product will, of necessity, be affordable; otherwise LMIC governments will simply not pay for it. "Traditionally," Charles Gore recalls, "MPP has not interfered in pricing. Our model is based on competition, and clearly we are potentially *giving this* to 15 companies around the world."(CG, emphasis added)

In contrast, researchers in South Africa who receive funding through the SAMRC must, under the terms of the funding, ensure that any "resulting products"—regardless of whether they target a PHEIC or not—will "be made available and accessible at an affordable price to people most in need within [LMICs]." Revealing differential treatment among participants in the programme, SAMRC-funded researchers and partners with products targeting a PHEIC have agreed to some form of pricing constraint whereas Afrigen has no such obligation unless and until it is in receipt of funding from SAMRC.

Significant questions about the enforceability of affordability clauses exist. Although they have included such clauses for "many, many years," one South African government official emphasizes, "it's really an aspirational clause" because "we haven't had to yet really test that."(XX) Other funders in the field of infectious diseases, notably CEPI and the Gates Foundation, are experimenting with various pricing commitments, such as "costs of manufacturing plus" a set percentage and tiered pricing (where products are priced lower in LMICs than HICs through confidential discounts) [55]. (JC) In contrast, MPP appears to be comfortable relying on free-market competition among LMIC-based manufacturers instead of imposing affordability clauses when it comes to products generated by virtue of participating in the mRNA programme.

In effect, the programme's approach reduces the pursuit of equitable access to the task of fostering more localized production. This is a logical step towards addressing local population health needs. But localized access is never guaranteed, particularly with initiatives that are expected to be "self-sustaining" businesses. Whether local manufacturers ultimately develop and sell their wares to local populations at an affordable price assumes, first, that those same manufacturers will maintain control over how their products are designed, where they will be launched and at what price; and, second, that local manufacturers' own business models and resource constraints will not compromise their pursuit of localized access and affordability. As we explain next, the web of transactional relationships that Afrigen and other programme participants have entered into may complicate that mission.

## Transactional R&D: Testing the limits of voluntary licensing

Under the dominant model of biopharmaceutical production, partnerships among the multiple actors engaged in the development of a biopharmaceutical product—from publicly funded labs to start-up companies, providers of research materials and equipment, contract research organizations (CROs), and large multinational manufacturers—tend to be *secret and*

*transactional* in nature [56–59]. Whether the aim is to secure research materials such as reagents or lipids, a license to use IP, assistance with recruiting participants for a clinical trial, or purchase outright a medium-sized company with a promising therapeutic candidate, agreements are generally actioned under conditions of confidentiality between two partners, with one typically acquiring the asset of interest from the other. Thus, ***enclosed, bilateral partnerships***—often short in duration—dominate biopharmaceutical R&D. More open and continuous knowledge-sharing arrangements through multilateral collaboration are, in contrast, relatively uncommon [33].

The original budget the South African consortium submitted to WHO was predicated on receiving technology transfer from an established mRNA manufacturer.(PT) Securing voluntary licenses to use IP is at the core of MPP's work and philosophy [38]. The organization "has no intention [. . .] of infringing any patents," MPP wrote while seeking funding from the Canadian government, "not least because MPP's key partners for licensing are pharmaceutical companies [52]." However, none of the HIC-based mRNA companies—Pfizer, BioNTech, Moderna, or CureVac—were interested in sharing their technology with the mRNA programme: "They didn't even want to talk."(PT) As a result, "the project turned into a green fields vaccine innovation," that is, "product development from nothing,"(PT) just as Bio-Manguinhos had proposed in 2021.(PN)

Looking to scale up rapidly but "wisely,"(CF) Afrigen began enhancing its own in-house capabilities where possible while outsourcing other elements of the manufacturing process. In order to make the drug substance and then formulate it into a product with the addition of an LNP, Afrigen purchased off the shelf a microfluidic device from Precision Nanosystems, a Canadian firm, to assist with the LNP encapsulation process.(CF) Fenner, Afrigen's scientific director, details how Afrigen overcame the key hurdle only to change plans in order to streamline costs for its LMIC-based programme partners down the road:

> [W]e knew that it was difficult to do LNP formulations and we saw all the skepticism and everything from everywhere else. [. . .] But for us we were like, 'Well, what was all the hype about really? We have been able to do it.' So we did use the Precision NanoSystems [PNI] machine, it's not that difficult to use. [. . .] And if you don't have access to lipids to do the actual formulation, the company themselves have a lipids mix which is proprietary to the company that they make available to their customers. And so that you can then formulate the mRNA into an LNP. [. . .] Having said that, we decided to not scale up on the [PNI] machine for the actual manufacturing. [T]he reason why we chose [another] machine is because that we thought that it is more simple to operate and that it has a lower running cost associated with it, which would be more appropriate for [LMICs].(CF)

While Fenner noted that Precision NanoSystems was acquired by Danaher Corporation after Afrigen began using its PNI machine, it was not clear whether Danaher's record of acquiring products and increasing prices, including for a TB diagnostic test [60,61], factored into Afrigen's decision to shift to another microfluidic device.

To demonstrate that AfriVac 2121 was 'non-inferior' to the Moderna and Pfizer/BioNTech's vaccines, it is necessary to perform preclinical testing in one or more animal models. The architects of the mRNA programme decided that aspect would be done by Xavier de Lamballerie's lab in Marseille, France, given that lab's experience using a hamster model to conduct SARS-CoV-2 challenge studies [62]. Marie-Paule Kieny, chair of the MPP's board, explains:

> [W]e wanted to have this in a center where the model has been validated internationally. So if Xavier de Lamballerie publishes that these results are equivalent, everybody will believe it.

If somebody in Afrigen is saying that it's the same, 'Uh-uh.' So, he's neutral, he's independent, he has no skin in the game. So he's testing the system. And now that we have this, so he has a lot of other studies to do, he will do neutralization of variants and so on and so forth, so this will be one package. And now he is also starting immunization of another batch of hamsters with the Afrigen product, the Moderna product, the Pfizer product, and this hamster will be challenged.(MPK)

When Lamballerie's preclinical studies of AfriVac 2121 are complete, the hamster model will be transferred to South Africa.(MPK,CF) As a result, "the local university [in Cape Town] is actually being capacitated. . .there's essentially a transfer of knowledge and protocols between the two so that in the future we would be able to do it in South Africa."(CF)

At each turn of the manufacturing process knowledge gaps are thus identified and addressed, often with the help of outsiders. Terblanche reports that Afrigen has at least nine different "cooperative research and development agreements" (CRADAs) at the "active implementation stage."(PT) [63–66] In some cases, the outsider's contribution is coupled with a commitment to assist Afrigen or another consortium member in gaining internal capacity, such as the hamster challenge model or performing GLP compliant toxicology studies, which Afrigen has outsourced to a "one stop shop" in India.(CF) In other instances, it is not clear whether the bilateral agreements will precipitate sustained collaboration around a shared set of goals. Meanwhile, mRNA programme partners are striking new deals and funding arrangements of their own. Bangladesh's Incepta, for example, has partnerships in the works or already in place with the University of Pennsylvania, Afrigen, Imperial College London, US NIH, and the Belgian company Quantoom.(MMA,MK)

It is notable that all of these bilateral CRADAs, funding agreements, and other contracts are the product of the programme's design. WHO and MPP, as the programme's architects, have chosen to place minimal constraints upon programme participants' ability to enter into bilateral agreements with external actors. The only stipulation under MPP's technology transfer template agreement is, if a partner of the mRNA programme obtains access to IP of a third party, the partner undertakes to "use reasonable efforts to negotiate a licence to MPP for such" third party IP. According to Terblanche, Afrigen has carried those access commitments through all of its CRADAs; where potential partners have balked at those terms, Afrigen has backed away from the deal.(PT) None of Afrigen's bilateral deals, nor those of programme partners, are publicly available, however.

Participating in the programme is a business opportunity. Serbia's Torlak Institute, for instance, has offered to sell reagents used during the production of influenza vaccines to other partners during the first programme-wide meeting held in Cape Town in April 2023.(LD) "I think this is the interesting part that we have," Bio-Manguinhos Missailidis explains, "you create a network that eventually there will be bilateral agreements within the network of people interested in some of our projects."

Outside actors engaged in the mRNA space have also increased their deal flow by virtue of their connections with the programme. According to Jose Castillo, head of Quantoom, which is known for its machines that automate an early part of the mRNA production process, already counts seven of the fifteen partners as its "customers" and is in active discussions with the other partners as well.(JC) Quantoom's contracts with the programme's partners moreover run deeper than simply selling its machines. Castillo recounts how he "knock[s] on the door talking about tech, but the contract I sign is a collaboration agreement"(JC). In return for assisting a partner to design an antigen against a pathogen of interest, Quantoom acquires a non-exclusive license to any project-related IP the partner in question generates.(JC) With agreements in place with many of the programme's participants, Quantoom stands to add

substantially to its IP assets, rendering it an attractive target for acquisition by a larger entity. Castillo's previous company, Artelis, was ultimately acquired by Danaher in 2015 [67].

The programme's architects are thus walking a fine line between trying to seed collaboration within and on the margins of the programme and trusting all involved to thread the commitments to IP access throughout that evolving web of relationships. Terblanche and Castillo appear steadfast in their commitment to the programme's stated objectives, yet cognizant of their respective organizations' vulnerability to market forces. Terblanche shares her thinking:

> I have a very strong, purpose driven, public health orientation. But I'm not stupid, I know my company needs to be financially viable to deliver on that promise. But greed is not my sin. Okay? I think that's the difference. But I can't tell you [. . .] what will be the next CEO's orientation? If Avacare [Afrigen's primary shareholder] dilute or sell[s] us [. . .] I have no control. So the only control I now have is agreements of care, which is public access. And these agreements will survive shareholders' ownership. That's the only thing I can do.(PT)

The decision to rely, to a significant extent, on private actors, banded together through CRADAs and other contracts, to build and share mRNA manufacturing capacities in LMIC settings is a signature feature of the programme. It is also reflective of the demonstrated preferences of its main architects, especially MPP, which has ascended in prominence in the sphere of global health as a result of the programme's development.

## Financialization's intermediaries: MPP as a rising 'power broker' in global health

A third defining feature of the biopharmaceutical sector today meriting comparison with the mRNA programme stems from the industry's **highly financialized** character. While the financialization of an industrial sector can manifest in several ways, the concept has generally been used to refer to the "increasing role of financial motives, financial markets, financial actors and financial institutions in the operation" of both domestic and international economies [68]. In the biopharmaceutical sector, the shift toward financialization is evident in the move by most major firms to become publicly traded on one or more stock exchanges (as opposed to family-owned businesses reliant solely on product sales for revenues) since the mid-twentieth century, the increasing importance assigned to maximizing shareholder value by financial actors with a controlling interest in many biopharmaceutical firms, and the growing reliance upon the tools of the financial sector, including mergers and acquisitions, and share buybacks and dividends, as the primary means to generate revenues [57,69,70].

The consequences of biopharmaceutical financialization are also several-fold. With financial companies, such as banks, venture capital firms, and asset management groups today often owning a controlling interest in any given biopharmaceutical firm, the strategic direction of those firms tends to become "more short-term oriented seeking to maximise immediate shareholder returns instead of making investments that look to the long-term health of the company [69]." Financialized biopharmaceutical companies may also increase prices for products already on the market to offset the cost of share buybacks and dividends, allocate greater resources towards marketing and advertising instead of R&D, and outsource R&D and manufacturing activities to countries, including LMICs, where labor costs are lower to the detriment of companies' in-house capabilities [69,70]. In fact, many biopharmaceutical firms actually outsource R&D and manufacturing activities to third-party CROs rather than perform the work in-house [71]. Outsourcing R&D has, in turn, created a space for a variety of intermediaries and consultants to develop business models of their own, connecting large

firms with CROs and other service suppliers in exchange for a fee, claims to IP, and reputational capital that flows from bridging various steps in the R&D process.

There is no indication that any of the mRNA programme's direct participants mirror the financialization that is evident among more established biopharmaceutical companies. Neither Afrigen, nor its primary shareholder Avacare, are publicly traded companies. The same is true of the other private companies partnering with the programme. None of the companies involved appear to be using the tools of finance such as share buybacks as a means to generate returns. While some inputs into the manufacturing process have been secured from outside entities, Afrigen, Biovac, and programme partners in other LMICs are invested in developing their R&D and manufacturing capacities in-house in an effort to increase local production in LMICs. The Belgium company Quantoom, which has locked in partnerships with most LMIC manufacturers in the programme, may be amassing an interest in each partner's IP. But Quantoom's machines, which automate the in vitro transcription step of mRNA production, seem to contribute meaningfully to the manufacturing process.

While MPP's position within the programme is not a symptom of financialization, the role that the foundation plays is analogous to the intermediaries that link together biopharmaceutical R&D and production supply chains. Like the industry's many intermediaries, MPP's presence simultaneously adds value to, and imposes a drain upon, the mRNA programme.

Through the programme's legal architecture, MPP has positioned itself as the central intermediary for technology transfer. Afrigen is the nominal hub for partners to receive training and it has provided a 3-day course about mRNA manufacturing to most participating manufacturers. However, it is MPP—at times, with the assistance of outside consultants (MMA, CN)—that has assumed the role of conducting site visits, assessing the needs and capabilities of each partner, and conveying the knowledge, data, and IP generated by the consortium. Awaiting for MPP to visit its facilities, representatives from one partner noted that receiving the transfer directly from Afrigen, as the producer of the drug substance, might be more efficient. But the partner cautions that its technology transfer agreement is with MPP, not Afrigen. As a result, "all the information comes from MPP." That is, "Afrigen to MPP and MPP to the spoke, [i.e.] to the technology transfer recipient."(XZ,XA)

Having MPP as a 'middleperson' is not the optimal way to provide technology transfer. (AK,EW,PN,PT) Typically, technology is transferred from one party, which has direct experience utilizing it, to another, through both sharing of hands-on know-how and detailed documentation such as 'Standard Operating Procedures.' Even in that two-party scenario mistakes occur, for example, when experimental protocols are not sufficiently delineated. Introducing an intermediary into the process, which lacks hands-on experience practicing the technology in question, increases the chances that the transfer will be unsuccessful. Nevertheless, over the course of 2022–2023, MPP created its own technology transfer unit, comprised of five individuals with doctoral and master degrees in chemical engineering and other fields as well as varying levels of experience in the pharmaceutical industry [72], to manage technology transfer within the mRNA programme. According to some interview participants, this has precipitated frustration with MPP's approach to technology transfer:

> The MPP is spending gobs of money on creating a group that is supposed to be doing tech transfer. I've worked for more than 30 years in the industry. You do not have a remote group that does tech transfer. If a group is going to do tech transfer, it needs to be in the facility that's sending the technology out. They need to be the detailed subject matter experts. So they seem to be building this empire that's going to do what, I don't know. And they seem to be micromanaging and want to be in charge of everything.(AK)

In that participant's view, "if [the programme] doesn't succeed, it's going to be because of that kind of dynamic, not because the science doesn't work."(AK) Altering that dynamic, moreover, requires tact:

> MPP now takes the funding for that [technology transfer] part and it goes to MPP because they have the team. So I don't want to be seen, now I want all the money for Afrigen. It's a [. . .] sensitive thing because people I think, they go sometimes, "because oh, you're Afrigen, you just want to dominate and control."(PT)

Afrigen will, according to Terblanche, assume responsibilities for the technology transfer in 2024. But whether the Cape Town company's people will be resourced to travel, train, and transfer technology to partners remains "under negotiations."(PT)

MPP clearly has the potential to add value to the programme in the IP domain. Its core expertise lies in evaluating patent landscapes for legal risk and negotiating licensing agreements with IP holders. MPP has compiled the patent landscape associated with mRNA technologies; the findings to date are worrisome. Of the nine LMICs linked to the programme through a partner for which patent data is available, patent applications have, according to MPP's dataset, increased markedly since a PHEIC was declared in January 2020 [23]. (see S2 Table) More than a third (56 of 159) of all patenting activity related to mRNA since 2006 in the 15 countries connected to the programme (concentrated mainly in South Africa, Brazil, India and Serbia) occurred between 2020 to 2022. Further, there are reportedly an increasing number of "use patents" (purporting to claim IP on the use of mRNA technology against a given disease, e.g., malaria) being filed in South Africa and other LMICs. WHO's Friede says such patents could block Afrigen and partners' R&D plans, "especially [. . .] in countries like South Africa where there's no substantive examination [of patent applications to ensure they meet standards of novelty, non-obviousness, and utility]."(MF)

Notably, MPP's work on the IP dimension of the programme does not extend to providing the South African consortium or other LMIC-based manufacturing partners with legal opinions about their respective "freedom to operate," (FTO) i.e., to conduct research and product development without undue risk of patent infringement litigation. The memorandum of understanding between WHO and MPP states that MPP will "provide IP analysis and commission [FTO] assessments for the Partners, as necessary [22]." However, the technology transfer agreements entered into with each partner refer only to providing "IP analysis. . .as practicable and appropriate" rather than supporting FTO assessments. MPP has reportedly provided funding for some partners to pay for FTO analyses by local law firms.(FL) Afrigen, however, has had to seek out independent legal advice using its own funding.(AK,PT) At least one partner is under the impression that the patent data that MPP has prepared and presented at programme meetings amounts to FTO guidance.(XZ,XA) But presenting a high-level summary of all the patent documents that exist is not the same as an in-depth interpretation of the scope of each patent granted in a jurisdiction—the core inquiry involved in crafting an FTO opinion.

The IP strategy behind the programme appears instead to be that the programme will meet incumbents' IP positions with IP of its own. Glaudina Loots of South Africa's department of science and innovation emphasises that the university scientists funded by the programme are keeping "the patent lawyers quite happy."(GL) At least two patent applications have been filed to date, one pertaining to antigen design and another on a novel LNP technology, with each listing inventors from the University of Witwatersrand, including Arbuthnot and chemist Charles de Koning. Arbuthnot contextualizes the importance of these new patent applications in light of the programme's goals of not just manufacturing a COVID-19 vaccine but also producing second generation mRNA technologies:

The lipid nanoparticle part of an mRNA vaccination technology is very, very important, so we've been doing a lot of work on that, and it's a particularly interesting project, actually, with some synthetic organic chemists here in Johannesburg at Wits as well, who are using a bio-renewable source for the manufacturer of the important ionizable lipids, they're called. [. . .] We're very excited about this because the lipids that are made to put into the vaccines that are used by, say, Moderna and Pfizer, are based on petroleum chemistry, so they're not bio-renewable products at all. If we are able to get products that can work as lipid nanoparticles, this is potentially something that could be quite big, and we're very excited and enthusiastic and working quite hard at trying to do that. That's an example of something that would enable us to have freedom to operate where we wouldn't have, wouldn't be dependent on the [IP] of anybody else.(PA)

While the experimental work to determine whether the bio-renewable lipid works is ongoing, "the medicines patent pool guys in Geneva," de Koning emphasises, "are telling us, 'You need to patent the stuff'" to ensure the hub has FTO.(CdK) Underscoring the potential strategic value of these patents in the event that the consortium or programme partners are confronted by a competitor down the road, WHO's Friede offers that "if that IP stands up to be really powerful, and we run into problems with certain companies, that IP might be a bargaining chip."(MF) In other words, a strategy of "defensive patenting" could help shield the programme from threats of patent infringement from outsiders while also sharing knowledge among participating manufacturers [73,74]. The success of that strategy will depend on whether the patents are ultimately granted.

In short, MPP's philosophy around IP, FTO, and contractual, negotiation-based solutions is imprinted on the programme; and, through its control over the programme, MPP has grown significantly in size and stature in the world of access to medicines. In the five years leading up to the pandemic, MPP's annual budget was in the range of CHF 5–6 million [75]. Since the pandemic began and the mRNA programme was launched, MPP's annual income has increased roughly fourfold (to CHF 23 and CHF 19 million in 2021 and 2022, respectively), the number of staff has jumped from 25 to 44, and their budget for external consultants has doubled [75]. Yet, it is not clear that MPP's presence and philosophy will work to the advantage of the programme's different participants. For instance, other go-to sources of funding in the field of infectious diseases, including the US NIH, the Gates Foundation and CEPI, have steered away from funding the programme as a whole. Gates recently provided USD 40 million spread between Cape Town-based Biovac, L'Institut Dakar in Senegal, and Quantoom, which has agreements with several programme partners [76]. CEPI has likewise supplied funding to four manufacturers taking part in the mRNA programme,(CN) including Indonesia's Bio-Farma [77]. None of these awards will flow through MPP, however. "I don't understand" why MPP and Gates are not working together, Quantoom's chief executive Castillo explains, and I have "that conversation with MPP and the same conversation with Gates, and not only with Gates, but with Mr. Gates."(JC)

Institutions engaged in the field of global health have long competed for resources, goodwill, and influence [78–81]. To the extent that there is fallout between actors vying for influence in Geneva, its impact upon the mRNA programme will not be uniform. Having expanded its portfolio of work to the realm of technology transfer, developed funding relationships with several HIC governments, opened a local chapter in South Africa, and launched a new strategic plan [82], MPP's place as a new "power broker" [81] within the sphere of global health appears assured. In contrast, the business at the heart of the programme, Afrigen, has thus far been unable to secure funding from Gates and other sources, and is, according to its head executive, increasingly "vulnerable" to financial strain.(PT)

### The sustainability question: Market-based governance

The fourth and final feature of the status quo is that control over the direction of biopharmaceutical production is concentrated in the hands of powerful, private actors that are, at bottom, ***governed by the market***. Established firms calculate the 'net present value' (NPV) of one disease target versus another, which systematically devalues diseases that are endemic in LMICs and thus offer lesser financial returns [83–85]. The entrance of the Gates Foundation and other philanthropic organizations has supplied new research funding for TB, HIV, malaria, and many other "neglected diseases." However, it is far from clear that these dominant organizations are prepared to rethink IP-intensive R&D practices or enforce access commitments in the service of public health [32,81,86,87]. They, too, are private actors, wielding significant influence over governments and entire scientific fields outside—and unaccountable to—the broader public.

Afrigen has targeted eleven potential diseases for mRNA product development. Unlike the standard approach to R&D, Terblanche suggests that the prospect of financial gains has not shaped the hub's priorities to date:

> Somebody asked me, [. . .] 'Your pipeline of 11. How did you get to it?' I said, 'I look at the unmet need, I look at the region. I look at whether there are vaccines or not, and I look whether the mRNA is suitable for it. And I look at whether we have access to antigens.' And they said, 'And what about the market?' I said, 'I've not included the market in my decision-making. I'm driving a need. And I'm not driving a profit, I'm driving a need, I'm driving sustainability.' (PT)

Multiple proposals focusing on Lassa fever, RSV, and other disease targets, have since been turned down by a variety of funders, leaving Terblanche to concede that her "hand will now be forced to prioritize" in light of market rewards.(PT)

The architects of the programme appear to have accepted Afrigen's precarity from the start, anticipating that the Cape Town firm might not survive in the hyper-competitive mRNA marketplace. The absence of pricing commitments parallel to those imposed upon programme partners or a commitment to allocate up to ten percent of "real-time production" capacity in the event of a PHEIC within the four corners of Afrigen's Grant Agreement with MPP [22] (*see also* S2 Table) could be interpreted as an incentive for Afrigen to commercialize its technology. Read in conjunction with the views of the programme's architects, the omission of these terms from the Grant Agreement likely suggests that the architects did not contemplate that Afrigen would ever generate mRNA products of its own. Recalling that some of the companies that took part in the influenza hub later shut down production, WHO's Friede estimates that if a handful of LMIC manufacturers manage to make mRNA vaccines, the newer programme will be an overall success.(MF) Charles Gore of MPP notes, "we are funding [Afrigen] to develop and then shift [the technology] out," but "they don't [yet] have a business model."(CG) Like any private enterprise engaged in biopharmaceutical innovation, MPP's Marie-Paule Kieny speculates that Afrigen will, in the end, probably yield to market forces:

> [W]ill the hub always be a necessity? I would argue no. [. . .] The hub is really there to establish a first platform and improvement, and to help with an early pipeline. After that we are fully aware that Afrigen is a private company, at one point they will try to find somebody to buy them out and to get the benefit. I don't know, what can we do? (MPK)

The near inevitability of Afrigen's exit in the eyes of those who designed the programme speaks to an underlying failure of imagination concerning how the mRNA programme is

governed. During the pandemic, calls for more "inclusive and decentralized" governance structures have grown [88] in order to shield initiatives such as the mRNA programme from the risks and constraints posed by dominant market actors. At a minimum, a more inclusive and decentralized structure would entail two key changes to the programme. First, representatives from participating LMICs capable of steering the programme's R&D towards local population would need to be directly involved in the programme's overall governance and day-to-day decision-making. Second, multiple actors would need to serve as regional mRNA hubs—as originally planned—in order to mitigate the risk that one organization's failure (or acquisition by an outside actor) might compromise the programme as a whole.

Instead, WHO and MPP internalized programme decision-making within two hand-picked committees, leaned on private actors like Afrigen to play crucial roles, preserved their discretion about what projects and partnerships to pursue, and limited input from LMIC governments and civil society during the programme's first two-plus years of operation. Members of the South African consortium, including scientists funded by the mRNA programme, also registered concerns about the lack of transparency surrounding the allocation of resources inside the programme. (ALW,ER)

The privatized approach to governance preserves the programme architects' control over the programme. The function of the "Scientific and Technical Review Committee" (or "STeRCO") is mainly to advise WHO and validate "high-level funding envelopes," including "the funding for the transfer of technology, the hamster model from Marseille to South Africa" and the purchase of large GMP equipment for preparation of mRNA for Afrigen.(MPK) For its part, the mRNA Scientific Advisory Committee (mSAC), which includes several notable experts with ties to industry, HIC governments, and academia, plays a more technical role, for example, evaluating the funding proposals submitted by South African researchers connected to the consortium. Reflecting their close working relationship, MPP's Kieny chairs the STeRCO while WHO's Friede chairs mSAC; the WHO is the secretariat for STeRCO while MPP supports mSAC's meetings and deliberations; and, the decisions by both serve to validate the use of funds from one source or the other (**Fig 2**). Missing altogether from this governance structure is any direct representation from LMIC governments whose needs and interests the programme is meant to serve.

Collectively, these choices reflect the programme's alignment with the dominant, market-drive approach to biopharmaceutical production. By design, the programme cannot stop Afrigen or other private companies embedded in the programme (e.g., Quantoom) from capitalizing upon the IP they amass through the programme or, if needed to satisfy their shareholders, sell their stake when the right offer materializes. Importantly, such a transaction would not mean that LMIC partners would be deprived of access to IP generated by Afrigen or other parts of the South African consortium. Similarly, a partner that patents an improvement of an mRNA technology that they receive from the programme cannot claim exclusive ownership over that new IP. Rather, the terms of the various agreements underpinning the programme pre-empt those possibilities. "[I]t's a self-executing grant-back obligation," MPP's head lawyer Park stresses, so "if a spoke partner invents something, patents it, the license automatically comes back to us."(CP) Notwithstanding this safeguard, however, the prospect of Afrigen exiting the space altogether or other parts of the consortium selling (on a non-exclusive basis) its IP assets to an entity based in an HIC, is in tension with the programme's stated goals of building local mRNA manufacturing capacity in LMICs.

## Discussion

The foregoing findings raise questions about the design choices embedded into the mRNA programme, how best to empower LMIC-based manufacturers, and what additional steps

**Fig 2. A schematic representation of the mRNA programme in principle versus in practice.** *Notes*: (1) This figure provides a schematic representation of the organizations and actors involved in the mRNA programme, and the inter-relationships between them. We contrast how the mRNA programme has been described in principle (Panel A) where Afrigen is described as the "hub" and manufacturers in LMICs are partners (or spokes), versus how the mRNA programme appears to work in practice based upon our findings (Panel B). (2) There are several important differences between Panel A and Panel B, including the fact that all of the technology transfer agreements are between partnering manufacturers and MPP (represented by double-arrowed lines). The only direct collaboration between Afrigen and a partner is with Bio-Manguinhos; the two organizations are in negotiations regarding a partnership focused on RSV (represented by the dashed line). Panel B also shows that all of the funding that has been secured for the programme flows through MPP (and WHO to a lesser extent). Members of the South African consortium as well as programme partners must negotiate access to funding from MPP, which, together with WHO, has delegated decision-making to the mSAC and STeRCo committees. Finally, Panel B also highlights other actors involved in the programme, including researchers at the University of Witwatersrand, the University of Cape Town, and North-West University (represented as Wits, UCT, and North-West, respectively), and the Belgium-based company, Quantoom, which has formed bilateral relationships with Afrigen and the majority of LMIC-based manufacturing partners. (3) While the figure's details derive from our research findings, they are not intended to provide a comprehensive picture of the mRNA programme. For example, recently the programme's architects have put into place a "Funders Forum" for countries and organizations that have provided financial support for the mRNA programme, and four new R&D consortia involving programme partners and outside actors were announced in March 2024 [89]. In order to limit the complexity of the figure, these new structures are not represented here.

need to be taken to ensure that this initiative enhances equitable access. We explore each in turn by way of conclusion.

## A facsimile of the status quo in biopharmaceutical production?

Assessed against the dominant mode of biopharmaceutical production, we find that the mRNA programme nuances—but does not substantially depart from—at least three of the four key features of the status quo. Funding sources wield considerable influence over the mRNA programme, and university-based researchers and publicly funded institutions have made critical contributions to manufacturing processes and second-generation mRNA technologies. Yet, apart from requiring IP sharing inside the programme, the architects of the programme opted not to stipulate terms of affordability upon partners' products (outside of the context of a PHEIC) on the strength of the assumption that market competition (either between the actors involved in the programme or with other mRNA manufacturers) will naturally yield that outcome. Further, the programme preserves every players' freedom to contract

with third parties. Thus, the programme's commitment to IP sharing rests on each organization's willingness to carry that condition through its various bilateral relationships, which have grown during the course of the programme. It is unclear if the programme provides oversight with respect to these bilateral transactions and, unlike the various legal agreements underpinning the programme, bilateral transactions between members of the South African consortium or manufacturing partners in other LMICs and third parties are also not publicly available. Finally, even though the entities involved are not embracing financialization in the manner of HIC-based biopharmaceutical companies, it appears that global market forces, rather than representatives of LMICs or local health needs, will ultimately decide what pathogens are prioritized for product development and whether Afrigen, the company at the heart of the programme, has a sustainable business model. To their credit, WHO and MPP are trying to seed collaboration within the programme, match-making participants around pathogens of shared interest and regional need [90]. But with trust among the diverse players still a work-in-progress, asymmetries in the programme's legal architecture in terms of who stands to earn IP royalties and where products generated with the help of the programme can be sold, and funding shortfalls on the horizon for the hub in Cape Town,(PT) it is far from clear that these collaborations within the programme will materialize before difficult decisions about licensing IP to HIC contexts will need to be made in the name of sustainability.

Rather than forming a collective enterprise, the relationships among local producers engaged in the programme are fragile and participants appear just as vulnerable to market forces as they were before the programme was created in 2021. This outcome is a by-product of how the programme has been designed, in particular, a number of choices made by MPP while constructing the programme. MPP's pursuit of a voluntary license from Moderna or another mRNA manufacturer arguably made sense initially given the state of the global health emergency when the programme began. However, MPP's preference for a voluntary deal with an HIC-based manufacturer also appears to have undercut Bio-Manguinhos' bid to become one of the mRNA hubs back in 2021. Bio-Manguinhos proposed building up end-to-end mRNA manufacturing capacity and 'South-to-South' technology transfer—a vision that the programme is now coming around to two-plus years later. The moment that a voluntary license from an established entity was not forthcoming, a deeper rethink of the programme's architecture and norms was needed. Instead, most of the features of status quo biopharmaceutical production have been carried through the mRNA programme and MPP continues to court voluntary agreements from Moderna and other HIC-based biopharmaceutical manufacturers [91].

## From remotely managed tech transfer to LMIC manufacturer empowerment

The mRNA programme's design reflects both the resource constraints that (in the absence of more funding) the programme must operate under as well as MPP's own institutional preferences around IP, competition, and solving access problems through voluntary (as opposed to mandatory) mechanisms. Without more funding from HIC governments, philanthropic organizations, or other sources, MPP was likely hard-pressed to demand that participating manufacturers agree to pricing constraints for mRNA products outside of a PHEIC. The money to make mRNA vaccines, in short, has to come from somewhere. Still, focusing exclusively on that economic reality obscures the high degree of control that the programme's architects have maintained over all decision-making, resource allocation, and technology transfer activities.

Representatives from LMICs, including South Africa, are largely excluded from the programme's governing structures. Likewise, civil society organizations from LMICs are not

meaningfully represented on the programme's STeRCo or mSAC committees. All of the funding for the programme sits in Geneva; only SteRCo and mSAC can decide if and when funding should be allocated. When Afrigen or another entity that is part of the programme is in need of additional funds, for instance, for the purposes of scaling up the next phase of technology transfer, or to pay for manufacturing equipment, it must engage in negotiations with MPP in order to access funding. Technology transfer, too, has to date been managed almost entirely by MPP's new technology transfer unit notwithstanding the inefficiencies that this approach may carry.

It remains to be seen whether MPP, which has ascended in the sphere of global health during the pandemic as a result of its role as the central power broker for the entire mRNA programme, will over time cede some of its control and take the steps necessary to truly empower LMIC manufacturers. The move to create R&D consortia as part of the programme, constructed around pathogens of shared interest, holds promise. Providing FTO opinions to LMIC manufacturers when needed, advising LMIC governments about when and how to invoke compulsory licenses, stepping back from micromanaging technology transfer among programme participants—all of those actions stand to re-distribute control and decision-making authority to the researchers, organizations, and governments in LMICs. Yet, compulsory licensing runs counter to MPP's way of doing business with established manufacturers and relinquishing control over the exchange of mRNA technology is in tension with the foundation's newfound mantle as the go-to facilitator of technology transfer within the sphere global health.

Far from unique to the mRNA programme, conflicts around power and control are commonly in play within multilateral initiatives. In theory, "multistakeholder models of governance promise greater participation of different stakeholders;" yet, in practice, "these models can also undermine the authority of intergovernmental organizations, while expanding opportunities for powerful private actors to exert influence over governing structures, and concentrating power among parties with less democratic accountability to poorer countries and populations [12]." MPP's rise has added to the "forum shifting", turf wars, and competition that preoccupies a number of global health institutions claiming, or already commanding, political capital from Geneva [78,81]. Too often in the past initiatives with the stated intention of combatting neglected diseases in LMICs have remained under the control of organizations rooted in HICs [92]. Claiming ownership over the mRNA programme, its architects see themselves as "giving" mRNA technology to LMIC partners.(MF,CG,MPK) Many partners in turn express gratitude to be involved but, as often occurs when power imbalances exist, remain subjugated by the gift [81]. If the mRNA programme's "decolonial aspirations" are to be realized [93], participating manufacturers in LMICs must be more empowered to collaborate South-to-South, build technological capacity, and generate mRNA interventions that are responsive to the health needs of, and affordable for, local populations.

## Translating investments in the mRNA programme into LMIC-centered scientific infrastructure, participatory rights, and power

The mRNA programme has in one sense already succeeded. Producing an mRNA COVID-19 candidate with limited assistance from outside actors in a six-month timeframe demonstrates the South African hub's technical capacity. From another perspective, the programme's success should not be construed exclusively in terms of product development, but its potential as a lasting safeguard of local production capacity in LMICs. Afrigen and the programme's partners are moving on to target other pathogens. Even if those efforts do not yield safe and effective mRNA vaccines against TB, RSV, malaria, and other priority diseases, the programme can

still have a lasting impact by generating an accessible body of scientific knowledge, tools, and people with the know-how to apply them—provided that those outputs remain within the reach of, and are responsive to, LMIC health needs.

The idea of linking public funding for biopharmaceutical innovation to public scientific infrastructure, participatory rights, and power gained renewed attention during COVID-19 [47,88,94–96]. However, most governments demanded little to nothing in terms of IP sharing, equitable access, and reasonable pricing commitments from biopharmaceutical companies in exchange for the massive public investments that were made towards mRNA vaccines, among other interventions [45,49,96]. The mRNA programme improves on this state of affairs by requiring sharing of IP among the South African universities and LMIC partners who have signed onto the programme. Yet it does not stop SAMRC-funded university labs from also licensing their IP to actors in HICs or LMIC partners from entering into bilateral deals that capitalize upon the IP and know-how they have gained through the programme. Quantoom, the Belgium-based company that is formally outside the programme and thus not subject to its IP sharing commtiments, has struck partnerships with the majority of the programme's manufacturers. Afrigen, for its part, can be sold to a third party if and when its primary shareholder decides that that course of action offers greater return than continued participation in the mRNA programme. All of these flexibilities that were built into the programme's design introduce a risk that the knowledge, tools, and know-how generated by the mRNA programme may be exploited or extracted by outside corporations that do not share the goal of improving equitable access in LMICs.

To mitigate that risk and ensure that the programme's outputs remain grounded in LMICs, a number of changes to the programme's legal architecture and governance structures are worthy of consideration. First, the programme's commitment to transparency should be extended throughout its operations. In particular, there must be greater transparency within the programme, such that Afrigen and other manufacturers can participate in, and have an understanding of, the decisions that are made by STeRCo and mSAC as well as the expectations of funders, including HIC governments. Similarly, the effort to make the programme's legal architecture transparent should encompass any bilateral deals that members of the South African consortium or LMIC manufacturing partners secure. Making those transactions transparent will help guard against the risk that the programme's collective IP and know-how will be captured by commercial outsiders. Second, representatives from LMICs taking part in the programme as well as civil society organizations from LMIC regions must have a stronger voice in the mRNA programme's decision-making structures. Running forums with LMICs and civil society after decisions have already been made by STeRCo or mSAC is insufficient. It distances the work of the programme from the very people it is intended to serve. Third, the prospect of Afrigen—the hub at the centre of the programme—being acquired by an outside entity, or dissolved altogether due to financial challenges, must be more proactively addressed. The architects of the mRNA programme contend that outcome is essentially inevitable. However, steps can be taken to ensure that the publicly funded R&D infrastructure, product leads, and scientific know-how that Afrigen has amassed are not lost in the event of a private acquisition or insolvency. In return for additional funding, WHO and MPP should require Afrigen to make all of its IP and know-how available to the other members of the programme. Further, WHO and MPP should engage with the government of South Africa to see if corporate restructuring, acquisition by the state, or some other strategic investment offers a means to preserve public control over the knowledge and assets that Afrigen, in collaboration with others inside and outside the consortium, has developed during the course of the programme.

Public investment intended to improve equitable access to health interventions, such as mRNA vaccines, requires policy innovation in the form of participatory rights for LMIC

communities and a protected stake in the scientific infrastructure and knowledge that that investment generates [96,97]. Revised with those fundamental goals in mind, the mRNA programme can position LMICs to lead where many HICs fell short in the context of COVID-19.

## Conclusion

The provision of a technological solution, including vaccines, is no safeguard for equitable access. Attention to social context and structural challenges is needed to realize technology's emancipatory potential. Our situational analysis of the WHO's technology transfer mRNA programme, including semi-structured interviews with 35 individuals involved to varying degrees in the programme, suggests that the needs and perspectives of LMICs are not sufficiently centered in the programme. Further, the architects of the programme are working within the existing system of biopharmaceutical production and, at the same time, preserving their own control over the programme's design and preferred measures meant to remedy shortfalls in equitable access to mRNA-based interventions. In particular, MPP continues to champion voluntary IP licensing as the optimal means to improve local production capacity in LMICs even though that mechanism did not attract collaboration from more established mRNA manufacturers in the context of COVID-19 and slowed adoption of a more transformative end-to-end approach to R&D and manufacturing. The technological outcomes of the mRNA programme remain uncertain. Absent significant reform and concerted effort to redistribute not just IP, but agency to LMIC actors, there is a significant risk that the programme, which is claimed by WHO and MPP as a collective effort to improve manufacturing capacity *in* LMICs *for* LMICs, will not solve the problem of equitable access to biopharmaceutical innovation.

## Supporting information

**S1 Letter. Response from Global Affairs Canada to Access to Information Request.**
(PDF)

**S1 Document. Disclosure Package from Global Affairs Canada.**
(PDF)

**S1 Table. Comparison of Legal Agreements Underlying the mRNA Programme.**
(DOCX)

**S2 Table. mRNA Related Patent Filings in Countries with Participants in the mRNA Programme.**
(DOCX)

## Acknowledgments

The authors express our gratitude to Morris Odeh, PhD student at Schulich School of Law, Dalhousie University, for research assistance on mRNA patent data in South Africa and LMICs; to the members of the IEAC (Fatima Hassan, Françoise Baylis, Jason Nickerson, Reshma Ramachandran, Srinivas Murthy, and Viviana Munoz) for their contributions and insightful feedback to this work and conversations that helped shape the form and substance of this manuscript and research project at large; to the Health Justice Initiative (HJI) for their support and engagement throughout this process; and to Els Torreele, Amy Maxmen, Melissa Barber, and Amy Kapczynski for their feedback, suggestions, and assistance during the research and writing process. Any errors or inconsistencies are solely the responsibility of the authors. Finally, we are especially grateful to the many individuals who took part in interviews

in connection with this research project. We hope our findings will play a constructive role as the work of the mRNA programme continues. All views in this paper are the authors and do not represent the views of the institutions they are affiliated with.

## Author Contributions

**Conceptualization:** Matthew Herder, Ximena Benavides.

**Data curation:** Matthew Herder, Ximena Benavides.

**Formal analysis:** Matthew Herder, Ximena Benavides.

**Funding acquisition:** Matthew Herder.

**Investigation:** Matthew Herder.

**Methodology:** Matthew Herder.

**Project administration:** Matthew Herder.

**Supervision:** Matthew Herder.

**Writing – original draft:** Matthew Herder, Ximena Benavides.

**Writing – review & editing:** Matthew Herder, Ximena Benavides.

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
