## [Decision Letter · Decision Letter 0]

1 Mar 2024

PGPH-D-24-00260

'Our project, your problem’: A case study of the WHO’s mRNA technology transfer programme in South Africa

Dear Prof. Herder,

Thank you for submitting your manuscript to PLOS Global Public Health. After careful consideration, we feel that it has merit but does not fully meet PLOS Global Public Health’s publication criteria as it currently stands. Therefore, we invite you to submit a revised version of the manuscript that addresses the specific points raised during the review process (see below). 

We look forward to receiving your revised manuscript.

Kind regards,

Roojin Habibi

Academic Editor

Journal Requirements:

b. If any authors received a salary from any of your funders, please state which authors and which funders.

If you did not receive any funding for this study, please simply state: “The authors received no specific funding for this work.

4. In the online submission form, you indicated that "While many of the interview participants took part in our study 'on the record', we did not ask if we could publicly share interview transcripts in their entirety. The data collected through semi-structured interviews with human participants will therefore only be made available upon request and subject to the approval of the university's research ethics board. All PLOS journals now require all data underlying the findings described in their manuscript to be freely available to other researchers, either 1. In a public repository, 2. Within the manuscript itself, or 3. Uploaded as supplementary information.

Reviewers' comments:

Reviewer's Responses to Questions

**Comments to the Author**

1. Does this manuscript meet PLOS Global Public Health’s publication criteria? Is the manuscript technically sound, and do the data support the conclusions? The manuscript must describe methodologically and ethically rigorous research with conclusions that are appropriately drawn based on the data presented.

Reviewer #1: Yes

Reviewer #2: Yes

2. Has the statistical analysis been performed appropriately and rigorously?

Reviewer #1: N/A

Reviewer #2: N/A

3. Have the authors made all data underlying the findings in their manuscript fully available (please refer to the Data Availability Statement at the start of the manuscript PDF file)?

Reviewer #1: Yes

Reviewer #2: Yes

4. Is the manuscript presented in an intelligible fashion and written in standard English?

Reviewer #1: Yes

Reviewer #2: Yes

5. Review Comments to the Author

Reviewer #1: This is an important paper that describes the attempt to circumvent the control that the large pharmaceutical companies currently have over R&D and access with respect to new technologies especially those based on mRNA. The paper does an in-depth evaluation of the potential and limitations of the Afrigen model and hopefully the analysis contained in this paper will lead to a re-evaluation of that model.

There are a number of places where I would like to see more detail provided or an expansion of the issues raised.

1. Was there an interview guide and if so, was it pilot tested? If there was an interview guide it should be included as an appendix.

2. Page 3: Is Afrigen a for-profit company? The authors answer this question in Figure 1 but I think that it should be mentioned at the start of the paper.

3. Was the WHO involved in helping to issue voluntary licenses, e.g., the AstraZeneca license to India's Serum Institute?

4. The authors might want to put some of their analysis about governance colonialism into the context of the governance of other similar initiatives and point out the prominence of organizations from the Global North - see Tucker et al. Science 2008;320:1016-1017.

5. Page 5: How did the authors get access to the documents that they analyzed? Did they have to sign some kind of confidentiality agreement? How many people extracted information from the documents? If more than one how were discrepancies resolved and if only one person what steps were taken to guard against biases?

6. Page 5: What role is the Canadian government playing in Afrigen's initiative? The authors have not mentioned Canadian involvement previously.

7. Page 11: Was there also opposition to C-TAP from HICs or just indifference?

8. Page 18: I'm confused here about programme partners' access to technology royalty free. In the preceding few paragraphs the authors only mention technology transfer to LMIC partners and from partners to MPP. Where do “programme partners” fit in? Are these the spokes?

9. Page 19: I'd like the authors to expand on their explanation about pricing. As I understand it, the South African hub has to provide any product at an affordable price to people in LMICs but the spokes in LMICs have no obligation to price products at an affordable price. Is my understanding correct? If so why would governments buy from the spokes if they can get the product at a lower price from the hub?

10. Page 31: The authors raise an interesting point that I'd like to see more discussion of. If it is known that Afrigen is interested in developing a vaccine for a disease will that discourage other companies from doing R&D in the same area? Will other companies only compete with Afrigen in financially lucrative disease markets?

Minor points

1. There are a number of minor copy-editing problems scattered throughout the manuscript that need to be corrected.

2. Page 19: One of the quotes in the last paragraph is not closed.

3. Page 21: Why isn’t Pfizer included in the list of HIC-based mRNA companies?

4. Page 29, second line from the bottom: There should be two years named but only one is named.

Reviewer #2: This is a useful paper, and will make a helpful contribution to the literature once published, with some relatively minor changes.

This is quite dense material. Some specific recommendations follow, but more broadly, it would be very helpful to have another diagram/figure (perhaps an organigram) outlining the relationships between the parties, and the governance structures.

- It is not always clear what conclusions are directly pulled from the interviews, and which are extrapolations derived by the authors. The first full paragraph on p.32 is one example of where this should be made clearer; the ref. to Danaher (flagged in more detail later) is another.

- The current uniform nomenclature for all anonymized sources is unhelpful. In truth, it may indeed matter whether the XX being quoted on a particular point is, say, a LMIC partner directly involved in this project, or an American academic whose involvement is unknown. I would defer to the authors/the editors on the precise format, but differentiating between all the anonymous voices (XX1, XX2 etc.) would be very helpful. I would consider this an important, and an obligatory, change.

- On the issue of anonymization, some of the descriptors are quite vague, particularly in the final category of global north scientists and other outside experts (e.g., “University researcher”). Appreciating that some discretion may be needed to preserve anonymity, slightly more information would be helpful in evaluating their contributions, especially when balanced against specific individuals whose credentials can be assessed by the reader comparatively easily. For instance, “University researcher” in what field?

- Given the (partial) reliance on Canadian government documents, a bit more discussion of both the disclosed material itself, and its broader implications (e.g., are there conditions govt funders could impose to address some of the concerns the authors raise?) would be welcome.

- A little more discussion of the origins of the “four paradigmatic factors” the authors have “abstracted from a review of literature” would be welcome. A reader seeking to use a similar framework would benefit from a stronger understanding of how it was derived. (If that derivation has already been published elsewhere, such a reference would probably suffice, although the reader could be directed there more clearly).

- On p.10, reference to “vaccine donation” from the global north could be clarified, as the difference between money to purchase doses and actual physical doses has proven relevant elsewhere in dissecting the COVID-19 response.

- The same is true for reference to funding “secured” and “committed” on p.15 (perhaps something more tangible like “delivered” as appropriate?).

- On p.11, it would be helpful to have further explanation and/or a reference for the mention of growing industry opposition to C-TAP (*appreciating that this mention is within the context of an interview).

- On page 16, some clarity is needed regarding Canada’s stipulation that funding be allocated to 4 countries. In numerous public statements, Canada has repeatedly made reference to providing funding to the Hub in Cape Town, with no reference to other spokes/countries. Particularly if, as the current wording suggests, no funding has actually flowed to South Africa, this deserves further explanation. Otherwise, more clarity would be helpful.

- A few minor typos throughout (e.g., a reference to “progamme” on p.28, a word missing in “a shared set commitments” on p.17, or “Médicins” rather than Médecins for MSF on p.8.)

- On p.19, I am curious about the authors’ insertion of “[had to]” within a quotation from an interview subject. It is not entirely clear to the reader that this insertion does not change the content or intention of the original speaker’s statement. It would be best if the authors checked their records and confirmed the speaker’s original meaning is indeed accurately conveyed (to be clear, it is possible that it may be, but it reads strangely).

- The text is particularly dense circa p.15-19, and this reader had to read this portion more than once. As noted above, a figure/diagram might help clarify the relevant relationships in a more readily comprehensible way.

- On p.22, it is unclear whether the speculation around Afrigen’s behaviour being influenced by Danaher’s actions around TB diagnostics stems from the interview subjects, or is purely that of the authors. This should be clarified.

- The last paragraph of p.25 seems a bit vague and speculative. The subsequent discussion flows better; it may be worth revisiting whatever point the authors are trying to make in this paragraph to better integrate it into the paper.

- On p.17, the authors refer to a PHEIC being declared in March 2020; the PHEIC was declared in January 2020, while March was the (more legally nebulous) pandemic declaration. It is not clear which the authors actually intend to use as their reference point.

- The discussion of MPP funding at the bottom of p.29 is a bit unclear. For instance, there is a reference to two figures, “respectively”, but only one year for those figures. It might also be useful to compare MPP funding pre COVID, given the pandemic impacted MPP’s activities even before the mRNA project, rather than solely pre/post mRNA.

- The discussion of different committees on p.32 is another example where a diagram might help illustrate governance.

- The last paragraph before the conclusion (p.33) needed to be read twice. Probably ok, but worth revisiting for clarity just in case.

- The abstract and the set-up reference colonial approaches, but until literally the last sentence of the paper, the direct link to the mRNA hub is not made clear. If this is really intended to be a key takeaway of the paper, it would withstand more discussion.

- As an aside… while perhaps not relevant to the outcome of the study itself, this reader is a bit curious how the ethics committee felt, in the era of Zoom and of climate change, about conducting 16 interviews on four different continents in person, rather than conducting them all virtually. If there was a clear justification for this bifurcation of interview formats, it may be worth including it.

- This reader thinks there is occasionally some strange and/or excessive comma usage, but defers to the journal editors in that regard.

6. PLOS authors have the option to publish the peer review history of their article (what does this mean?). If published, this will include your full peer review and any attached files.

**Do you want your identity to be public for this peer review?** For information about this choice, including consent withdrawal, please see our Privacy Policy.

Reviewer #1: **Yes: **Joel Lexchin

Reviewer #2: No

---

## [Decision Letter · Decision Letter 1]

12 Jul 2024

PGPH-D-24-00260R1

'Our project, your problem’: A case study of the WHO’s mRNA technology transfer programme in South Africa

Dear Dr. Herder,

Thank you very much for submitting your revised manuscript to PLOS Global Public Health and responding to reviewer comments. We have received further minor feedback from the reviewers regarding revisions made. Overall, we feel that you and your co-author have done an excellent job of responding to reviewer comments. To the extent that it is feasible, we are sharing further comments received from reviewers. They are minor so I hope that they can be addressed with minimum time and effort. For any that you deem unnecessary to address, please feel free to leave out and briefly indicate to us why you chose not to address the feedback.

We look forward to receiving your revised manuscript.

Kind regards,

Roojin Habibi

Academic Editor

Journal Requirements:

Reviewers' comments:

Reviewer's Responses to Questions

**Comments to the Author**

1. If the authors have adequately addressed your comments raised in a previous round of review and you feel that this manuscript is now acceptable for publication, you may indicate that here to bypass the “Comments to the Author” section, enter your conflict of interest statement in the “Confidential to Editor” section, and submit your "Accept" recommendation.

Reviewer #1: (No Response)

Reviewer #2: All comments have been addressed

2. Does this manuscript meet PLOS Global Public Health’s publication criteria? Is the manuscript technically sound, and do the data support the conclusions? The manuscript must describe methodologically and ethically rigorous research with conclusions that are appropriately drawn based on the data presented.

Reviewer #1: Yes

Reviewer #2: Yes

3. Has the statistical analysis been performed appropriately and rigorously?

Reviewer #1: N/A

Reviewer #2: N/A

4. Have the authors made all data underlying the findings in their manuscript fully available (please refer to the Data Availability Statement at the start of the manuscript PDF file)?

Reviewer #1: Yes

Reviewer #2: Yes

5. Is the manuscript presented in an intelligible fashion and written in standard English?

Reviewer #1: Yes

Reviewer #2: Yes

6. Review Comments to the Author

Reviewer #1: The authors’ revisions have dealt with my initial concerns. I have two further suggestions for areas to comment on (although these are not essential to include) and there are a number of minor issues.

Issues to consider:

Page 27: Do the authors want to talk about how financialization results in turning patents and IP into assets to be bought and sold thereby increasing the price of any eventual product?

Page 33: Do the authors want to get into a discussion of negative aspects of involving the BMGF in funding vaccine projects - see The Bill Gates Problem by Tim Schwab?

Minor issues:

Page 10: The authors should make it clear that contributions to C-TAP were to be made on a voluntary basis.

Page 15: It should be "definition" not "defining".

Page 16, second paragraph: Are the authors talking about all phases of biopharmaceutical R&D or just basic research being publicly funded?

Page 25, line 5: It should be "member" not "member's".

Page 34: Is it Afrigen or Afrigen's principal shareholder Avacare that has been having trouble securing additional funding?

Page 35, 8th line from bottom: Insert "of" between "handful" and "LMIC".

Page 36: The sentence starting "At a minimum..." is very long and difficult to follow. The authors should break it up.

Reviewer #2: Good paper, all major concerns addressed. Two exceedingly minor comments. First, the section "The Sustainability Question: Market-Based Governance" in particular should be re-read with an eye to grammatical typos, e.g., "Friede estimates that if a handful [of] LMIC manufacturers"; "to validate the use [of] funds from one source"; "in the service of the public health" etc. Second, Figure 2, especially the second portion, is very difficult to read at 100% size; if it's anticipated anyone would ever read it without being able to zoom a PDF, it might be worth making it larger.

7. PLOS authors have the option to publish the peer review history of their article (what does this mean?). If published, this will include your full peer review and any attached files.

**Do you want your identity to be public for this peer review?** For information about this choice, including consent withdrawal, please see our Privacy Policy.

Reviewer #1: **Yes: **Joel Lexchin

Reviewer #2: No

---

## [Editor Report · Decision Letter 2]

23 Aug 2024

'Our project, your problem’: A case study of the WHO’s mRNA technology transfer programme in South Africa

PGPH-D-24-00260R2

Dear Mr. Herder,

We are pleased to inform you that your manuscript ''Our project, your problem’: A case study of the WHO’s mRNA technology transfer programme in South Africa' has been provisionally accepted for publication in PLOS Global Public Health.

Best regards,

Roojin Habibi

Academic Editor
